# Trends in Sales and Industry Perspectives of Package Sizes of Carbonates and Confectionery Products

**DOI:** 10.3390/foods10051071

**Published:** 2021-05-12

**Authors:** Chloe Jensen, Kirsten Fang, Amanda Grech, Anna Rangan

**Affiliations:** Nutrition and Dietetics, School of Life and Environmental Sciences, Charles Perkins Centre, The University of Sydney, Sydney, NSW 2006, Australia; cjen7207@uni.sydney.edu.au (C.J.); kfan0424@uni.sydney.edu.au (K.F.); Amanda.grech@sydney.edu.au (A.G.)

**Keywords:** package size, Euromonitor, carbonates, confectionery, monitoring, public health, food industry

## Abstract

Discretionary food package sizes are an important environmental cue that can affect the amount of food consumed. The aim of this study was to determine sales trends and reported food industry perspectives for changing food package sizes of carbonates and confectionery between 2005 and 2019. Changes in package sizes of carbonates and confectionery were investigated in Australia, the USA, Canada, and the UK. Sales data (units per capita and compound annual growth rate between 2005 and 2019) were extracted from the Euromonitor database. Qualitative data (market research reports) on industry perspectives on package size changes were extracted from industry and marketing databases. Carbonate sales data showed increased growth of smaller package sizes (<300 mL) and a decrease in sales of larger package sizes (≥2000 mL) in all four countries. In contrast, confectionery sales data showed no consistent trends across the selected countries. No growth was observed for smaller confectionery package sizes but an increase in growth of larger package sizes (50–99 g, >100 g), including share packages, was observed in Australia. Qualitative data (*n* = 92 articles) revealed key reasons identified by industry for changes in package size related to consumer health awareness, portion size control, convenience, market growth, and government or industry initiatives. Monitoring of discretionary food package sizes provides additional insights into consumers’ food environment.

## 1. Introduction

Over the past 30 years, portion sizes of many foods and beverages have increased, in particular nutrient-poor, energy-dense foods [1,2]. People consistently consume more food when offered larger sized portions, packages, or dishware (tableware) than when offered smaller-sized versions [3]. This ‘portion size effect’ has been observed in children and adults, males and females, and across those with different body weights, levels of dietary restraint, and susceptibility to hunger [3]. Snack foods and foods with a high energy density and/or low nutrient profile are particularly susceptible, likely due to their high palatability [3,4]. As there is little compensation in energy intake at subsequent meals, consuming larger portion sizes leads to higher energy intakes and increased risk of overweight and obesity [3,5].

The consumption of large portion sizes of energy-dense, nutrient-poor foods is particularly concerning [6,7]. In many countries, the consumption of such foods high in added sugar is excessive; for example, in Australia, sugar sweetened beverages (SSBs) and confectionery are leading contributors of added sugars and/or saturated fat intake [8]. Similarly, a 2010 report on American diet and health revealed that SSBs contributed 36% of total added sugars and confectionery 6% [9], and, in the UK, confectionery and SSBs were the main contributors to free sugar intake in adults in 2016 [10]. In Canada, SSBs were the main beverage contributor to energy intake in 2015 [11]. To reduce population intakes of these foods, targeting the portion and package sizes of these products has been proposed as a potential strategy. A Cochrane review [3] and other recent reviews [5,12,13,14] suggest that policies and practices that reduce the size, availability, and appeal of larger sized portions and packages can contribute to meaningful reductions in the quantities of food people select and consume immediately and in the short term. Environmental cues that promote the selection of smaller servings could be a valuable strategy to reduce portion sizes. Thus, a range of package sizes, with more options towards the lower end, is an important consideration to help redefine a new ‘normal’ [15,16,17] and guide consumers to reduce their energy intake at one occasion. Few studies have been conducted that examine the response of the food industry to the package sizes of discretionary foods.

The use of marketing databases such as Euromonitor International (Euromonitor) can provide valuable data and market research for changes in package sizes over time. The use of such databases has thus far been quite limited in public health nutrition research, likely due to the cost of access and publication restrictions. For this study we selected carbonates and confectionery as examples of packaged food products that are energy-dense, nutrient-poor, commonly consumed in high-income countries, available in a variety of package sizes, and well-defined as food product categories in Euromonitor. Four high-income countries—Australia, the United States of America (USA), Canada, and the United Kingdome (UK)—were chosen to compare and contrast any changes over time. It is a legal requirement that the net weight of the food is provided on all packaged foods in all four countries. Quantifying how package sizes of unhealthy food and beverages have changed over time helps to develop a better understanding of the food environment and how the food industry can influence the population’s diet. Determining the principals that motivate industry to make changes to package sizes can assist in understanding the barriers and enablers that influence package size reduction and inflation. The aims of our study were firstly to examine the trends in sales of varying food package sizes over the last 15 years for carbonates and confectionery in four high-income countries and secondly to investigate the industry-reported reasons for these trends.

## 2. Methods

### 2.1. Sales Trends According to Package Size: Carbonates and Confectionery

Sales data for carbonates and confectionery were obtained for analysis from the Euromonitor Passport Global Market Information Database, 2019 Edition (Euromonitor). This market research database contains data from multiple primary and secondary sources, including company financial reports, store audits, official government statistics, and data from industry bodies [18]; however, the exact data sources for carbonates and confectionery are not available. ‘Carbonates’ include sweetened, non-alcoholic drinks containing carbon dioxide, both regular and low calorie and naturally and artificially sweetened, but exclude carbonated water, tea drinks, and energy drinks; while ‘confectionery’ includes chocolate confectionery, gum, and sugar confectionery. Sales data for units of packages (or packs) were obtained from 2005 to 2019 for four countries: Australia, the USA, Canada, and the UK. The search strategy is illustrated in Appendix A.

Sales data, including package unit sales and compound annual growth rate (CAGR), were exported from Euromonitor into Microsoft Excel for the years and countries of interest. Package unit sales for carbonates were classified as total sales and separately for retail and foodservice sales. Only retail sales were available for confectionery. Retail sales were defined as sales through establishments engaged in sales of goods, including supermarkets, convenience stores, department stores, and grocery retailers. Foodservice sales were defined as sales to foodservice establishments, such as restaurants, cafes, bars, fast food outlets, home delivery and takeaway services, self-service cafeterias, kiosks, and street stalls [18] (Appendix A). The CAGR is defined as the annual average growth rate, expressed in percentage terms, for the selected forecast period.

Sales data were organised according to package size and year, with package sizes placed into size bands and years placed into five-year bands to explore sales trends over time. Five package size bands were chosen for carbonates: <300 mL, 300–399 mL, 400–999 mL, 1000–1999 mL, and ≥2000 mL, and four package size bands were chosen for confectionery: <25 g, 25–49 g, 50–99 g, and ≥100 g. The five-year bands were 2005–2009, 2010–2014, and 2015–2019. Package unit sales (total and retail and foodservice sales where available) were summed according to package size and year band. The package unit sales data were converted to per capita sales using population statistics available on Euromonitor. All sales data were graphed by country for unit sales per capita for each five-year band and for the per capita CAGR between 2005 and 2019. Growth was considered as positive if there was ≥ +1% change; negative if there was ≥−1% change; and stable if there was <1% change over 15 years.

### 2.2. Food Industry Perspectives on Package Size: Carbonates and Confectionery

To document the food industry perspectives on package size changes over time, a systematic approach was adopted by applying the Preferred Reporting Items for Systematic Reviews and Meta-Analyses Extension for Scoping Reviews (PRISMA-ScR) guidelines [19].

Sources were considered eligible if they were articles such as market reports and trade publications, had a title or summary containing information on changes to package sizing for carbonates or confectionery, were obtained from Australia, the USA, Canada, or the UK, and were published between 2005 and 2020. Sources were excluded if changes in package sizes were not clearly described. Euromonitor, IBISWorld, and the internet search engine Google were searched. Access to Euromonitor was available from 2005–2020 and to IBISWorld from 2019–2020. IBISWorld is an industry market research database comprising reports written by expert analysts utilizing worldwide economic, demographic, and market data [20]. Google uses an authority-based algorithm that displays ranked results by relative importance depending on the linked domain.

The search strategy used to extract industry reports from the Euromonitor database is described in Appendix A. A systematic key word search to obtain additional qualitative data was performed on Euromonitor, IBISWorld, and Google (Appendix A). The first 100 results from Google were analysed, as per Dumas et al. [21]. All searches were conducted on 16 April 2020. Two reviewers (C.J. and K.F.) independently sourced and assessed the eligibility of publications identified by the search strategies. The screening process involved title and abstract or introduction review followed by full text appraisal. Disagreements over inclusion or exclusion were resolved through discussion with a third party (A.G. and A.R.). A full list of sources and websites is available on request.

The extraction and charting of industry report data were performed in duplicate by two independent reviewers using a customised template designed for this study. Data items included country of report, year of publication, title, article type (e.g., opinion article, briefings), direction of change in package size (increase or decrease), specific changes in package size, and the reported reasons for the change. Any discrepancies in judgement were discussed and consensus reached on all occasions.

## 3. Results

### 3.1. Sales Trends According to Package Size: Carbonates (Retail and Foodservice)

Total per capita unit sales of carbonates were highest in the USA, followed by Canada, Australia, and the UK, between 2005 and 2019 (Figure 1), with the most popular package size being 300–399 mL in these countries. Total per capita unit sales decreased during this time period in Australia, the USA, and Canada but remained relatively stable in the UK. With regard to package sizes, positive growth (per capita CAGR) was shown for <300 mL carbonates, while growth for ≥2000 mL carbonates decreased in all four countries between 2005 and 2019 (Figure 2).

For all countries, retail unit sales accounted for the largest proportion of total carbonate unit sales (70–92%) and showed similar trends to total carbonate unit sales (Appendix A). Total foodservice unit sales decreased between 2005 and 2019 in Australia, Canada, and the UK for all package sizes. The most popular sizes in foodservice were <300 mL in the UK, 300–399 mL in Australia and Canada, and 400–999 mL in the USA (Appendix A).

### 3.2. Sales Trends According to Package Size: Confectionery (Retail Only)

Per capita unit sales of confectionery were highest in the UK, followed by the USA, Canada, and Australia. Between 2005 and 2019, total per capita unit sales decreased in the USA, whereas unit sales remained relatively stable in Australia, Canada, and the UK (Figure 3). No consistent pattern was found in terms of unit sales of package sizes during this time across the four countries. Between 2005 and 2019, growth (per capita CAGR) of small package sizes (<25 and 25–49 g) decreased or remained relatively stable in all countries (Figure 4). Growth of larger package sizes (50–99 g and >100 g) was observed in Australia but not in other countries where sales remained stable (USA and Canada) or decreased (UK).

### 3.3. Food Industry Perspectives on Package Size: Carbonates and Confectionery

The initial search of the electronic databases identified 451 articles (market reports and trade publications) and, after removal of duplicates, resulted in 408 articles (Figure 5). Following screening of the titles and abstracts/introductions, a further 289 articles were excluded. Full texts were retrieved for 119 articles for detailed evaluation against the eligibility criteria and a total of 92 articles (carbonates, *n* = 51; confectionery, *n* = 39; and carbonates and confectionery, *n* = 2) were included for qualitative synthesis. Articles came from a variety of sources, such as ConfectioneryNews.com, Forbes, Beveragedaily.com, Euromonitor research reports, and direct from company websites. Most of the included articles originated from the UK (particularly confectionery), the USA, or Australia with fewer from Canada. The majority of articles were published after 2013.

#### 3.3.1. Carbonates

All included studies (*n* = 53) were characterised according to the direction of package size change and the reason for this change as identified by industry (Table 1). For carbonates, 46 publications identified a decrease, two studies identified an increase, and five found a mixture of increasing and decreasing package sizes (Appendix A for full details of articles). The main reasons reported for decreasing package sizes were increased consumer health consciousness (*n* = 30) regarding sugar and calorie content, convenience (*n* = 17), portion control (*n* = 14), market growth (*n* = 13), and innovation (*n* = 12). The reasons offered for increasing package sizes included value for money (*n* = 4), expanding product line (*n* = 2), market growth (*n* = 1), and demand from certain age groups (*n* = 1). This latter demand for larger sizes originated from 14–18 year old males in Australia [22].

#### 3.3.2. Confectionery

Out of 41 articles on confectionery package sizes, 24 identified a decreasing trend, 7 found an increase, and 10 showed a mix of both (see Table 2 and Appendix A for full details of articles). The main reasons reported for reducing package sizes were portion control (*n* = 18), increased health consciousness of consumers (*n* = 16), allowance for small indulgences (*n* = 14), to reduce manufacturing costs (*n* = 10), and to increase affordability (*n* = 8) (Table 2). These reductions in package sizes were due to either small changes to the original package size or the introduction of new smaller pack/pouch sizes. Incremental reductions of 10–20% were reported in several articles [23,24,25,26,27,28,29,30,31]. The introduction of new smaller sizes was also reported, such as new Cadbury bars at 35 g and Hershey Sticks at 11 g (60 calories) [32,33]. Some companies introduced portion-controlled package sizes and others introduced new high-end products, or premiumisation, reportedly as a response to consumers placing more value on quality rather than quantity. Some manufacturers have altered their larger single-serve confectionery bars (e.g., king size) into two or more smaller sizes while maintaining the overall weight of the product in an effort to prompt sharing, aid in portion control, and encourage impulse purchases.

Another trend observed in all countries was an increase in the release of larger sized packs, or ‘share packs’, (15 out of 16 articles reporting increases in package sizes concerned share packs), with sizes varying from 110–440 g. The main reasons documented for this change by the food industry were a rise in sharing trends (*n* = 14) and value for money (*n* = 7). Additionally, a decrease in the size of confectionery pieces inside the larger share packs, known as ‘miniaturisation’ of countlines (or mini-bite-sized products), was noted, reportedly to further prompt sharing trends and aid in portion control.

## 4. Discussion

Using a systematic methodology, this study assessed the sales trends of carbonates and confectionery according to package size over a 15-year period along with industry perspectives on package size changes, in four high-income countries. Per capita unit sales data showed that consumers are increasing their purchases of smaller sized carbonates (<300 mL) and decreasing their purchases of larger sized carbonates (≥2000 mL). Industry reports mirrored this trend, citing consumer concerns over sugar and calorie content as reasons for the move towards smaller, portion-controlled sizes. However, unlike carbonates, confectionery sales data showed no consistent trends in package sizes across the selected countries over this time period. Sales of small package sizes (<25 g and 25–49 g) decreased in Australia and Canada whereas larger package sizes (>100 g) increased in Australia but remained relatively stable in the USA, Canada, and the UK. Industry reports described both decreases and increases in confectionery packaging for various reasons; single-serve confectionery sizes have gradually reduced in size, while larger share packs are gaining popularity.

An increased focus on reducing population sugar and energy intakes to curb obesity rates has led to increasing pressure on soft drink and confectionery manufacturers to act responsibly. Initiatives such as reformulation, public education campaigns, and the implementation of a tax on sugary drinks have been trialled and/or implemented in various countries. For example, in the UK, a Soft Drinks Industry Levy (SIDL) has been introduced, which is a two-tiered industry levy (5–8% sugar content at 18 p/L and >8% sugar content at 24 p/L) intended to encourage reformulation, encourage a shift to lower sugar alternatives, and a reduction in package sizes [34]. This levy was an important reason for the decrease of carbonate package sizes reported by industry in our study. However, as the tax was introduced relatively recently, in 2018, we were unable to detect the impact on sales of smaller package sizes of carbonates. A UK study that compared carbonate sizes before and after the levy in leading/major supermarkets found that the SIDL led to an overall reduction of sugar in soft drinks, mostly due to reformulation, with little change in product size, with the exception of the small category of own-brand drinks [34]. There are currently no sugar taxes in Australia, Canada [35,36], and much of the USA (with the exception of a few cities and districts) but all countries have implemented educational programs that encourage decreased consumption of SSBs in various populations. As consumers are progressively opting to purchase smaller sized carbonates, some of these initiatives may be having an impact.

The majority of the carbonate sales were retail sales, with only a small proportion from foodservice outlets, and our findings indicated a decrease in per capita unit sales of carbonates in foodservices in Australia, Canada, and the UK but not in the USA. However, in foodservices carbonates can be sold in packages (cans or bottles) or in cups using post-mix dispensers. The latter measure is unavailable in the Euromonitor database. The most popular package sizes were similar for both retail and foodservice in Australia and Canada (300–399 mL) but differed in the USA (400–999 mL was more popular in foodservice) and the UK (<300 mL was more popular in foodservice). Reasons for these differences can only be speculated but are likely due to different target population and cultural expectations.

Public health initiatives, such as labelling and promoting healthier beverages, and price increases on sugary drinks in food outlets have shown some success in reducing sugar-sweetened beverage consumption [37]. A proposed ban on serving sizes greater than 16 ounces (470 mL) in New York foodservice establishments was not implemented due to beverage industry, business, and consumer opposition [38]. Altering portion sizes as a nudging intervention to reduce sugar-sweetened beverage consumption may be a more acceptable approach, although country-specific factors need to be considered, as one study found that consumers in the US may be more resistant than those in the UK [39]. Voluntary industry initiatives include the Balance Calories Initiative, which has led the top three American beverage companies to commit to promoting the use of smaller package sizes. An interim 2020 report found some growth among smaller containers of caloric beverages but this was offset by shifts from medium-sized to larger containers [40].

Confectionery sales data trends showed mixed results for package sizes by country. In Australia, growth in larger package sizes (>100 g), which typically include share packs, was observed. Industry reports confirmed that larger share packs of chocolates and sugar confectionery are gaining popularity. Resealable share packs are growing and are marketed as ‘permissible snacking, encouraging consumers to share and enjoy the experience with friends and family’. Interestingly, it was found to be common for the size of individual confectionery portions within the larger sized share packs to decrease over time [41,42,43,44]. This was reportedly to ‘help consumers with portion control, as these packs are a way for consumers to access portions smaller than the individually bought single-serve confectionery’ [43]. Whilst these share packs have the potential to help consumers with portion control, there is currently little evidence to support this. A study that investigated how much of a ‘portion-controlled’, two-piece, king-size confectionery bar consumers would eat found that, despite the bar being portion controlled, consumers ate both pieces at one time [45].

In contrast to the growth in share packs, our review also found that manufacturers decreased the size of some single-serve confectionery packages. Incremental reductions of 10–20% were reported in several articles [23,24,25,26,27,28,29,30,31], exemplified by a Euromonitor analyst blog in 2015 that commented that Mars, Mondelez Hershey, and Nestlé had reduced the size of their countlines by 10–20%, usually without reducing the unit price of the products [25]. This is commonly referred to as shrinkflation [24,25,27,28,43,44]. The most common reasons reported by the food industry for reducing confectionery package sizes were consumer health awareness, portion control, and allowing consumers ‘to enjoy a permissible indulgence with less guilt’, as well as to increase profit levels and achieve market growth [46,47,48,49]. However, the higher price per unit weight may deter consumers from purchasing these smaller sizes, as value for money is reportedly of great importance to consumers [23].

Several voluntary industry initiatives targeting confectionery have been implemented. In Australia and the UK, ‘Be Treatwise’, launched in 2006, aims to ‘help consumers understand the role of confectionery as a treat food, and as a reminder to be responsible with portion sizes’ [50]. In the USA and Canada, ‘Always A Treat’ aims to provide clear calorie labelling and more choice in smaller packages [51]. Neither of these initiatives was identified as part of the scoping review and their effectiveness is yet to be determined, with our review finding no evidence of increased sales of smaller confectionery packs over the past 15 years.

This study included a variety of data sources, qualitative and quantitative, and investigated a range of databases which provided good coverage of food industry interests in changing package sizes. The selected countries were large, high-income countries and therefore the trends found represent a large proportion of the developed world. All data were double screened, allowing for greater quality and reliability of results. Although the study provided good coverage of data for each of the countries, only a few industry articles were published between 2005 and 2019 in the Euromonitor and IBIS databases and Google search engine. Limitations of the study included the lack of separation of sugar-sweetened versus intensely-sweetened beverages in the Euromonitor sales data by package size; the fact that confectionery sales from foodservice outlets were unavailable; and the inability to distinguish between sales of share packs and chocolate bars, as both were all categorised in the ≥100 g size band. Additionally, limited data were available on the consumer characteristics of those purchasing smaller versus larger package sizes. The package sizes used in our analysis were summarized as band widths and using individual package sizes may detect more subtle changes over time.

## 5. Conclusions

Monitoring package sizes of discretionary foods and beverages, and recognising the reasons for modifying, and particularly downsizing, package sizes, is important to improve the food supply, assist consumers in eating healthier diets, and reduce levels of overweight and obesity. Our study presents a novel insight regarding sales trends of discretionary food package sizes, finding growth in smaller package sizes in carbonates but more diverse findings for confectionery. Promisingly, health consciousness was found to be the primary driver for reductions in both confectionary and SSBs. However, the food industry is also motivated to increase profitability and employs a variety of strategies, such as using both larger and smaller package sizes, to increase sales to consumers who are motivated by factors other than health, such as value for money. As smaller packages also drive sales and potentially allow new indulgences that may not have occurred with larger sizes perceived to be unhealthy, further research into potential unintended risks associated with consumption of smaller package sizes is warranted.

## Figures and Tables

**Figure 1 foods-10-01071-f001:**
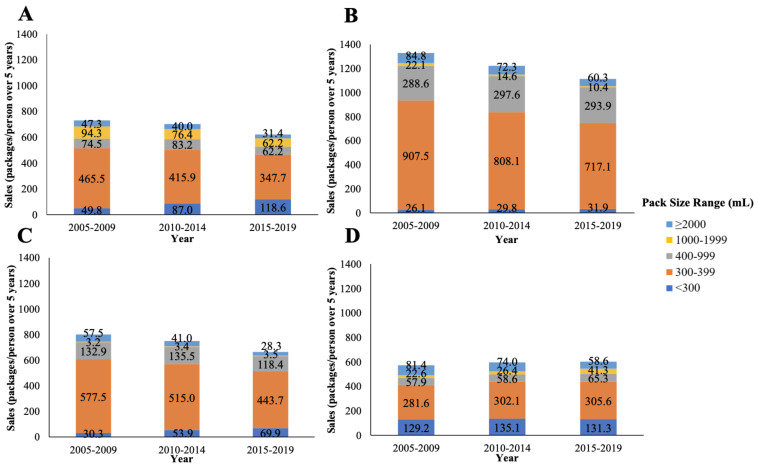
Total package sales per capita, including retail and foodservice sales, over five-year periods from 2005 to 2019 in carbonates according to package size band in (**A**) Australia, (**B**) the USA, (**C**) Canada, and (**D**) the UK.

**Figure 2 foods-10-01071-f002:**
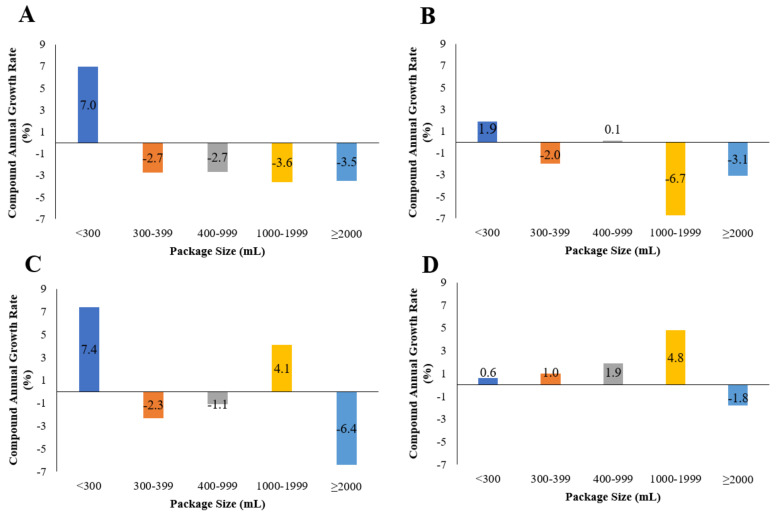
Compound annual growth rate of total package per capita sales, including retail and foodservice sales, of carbonates for 2005 to 2019 according to package size band in (**A**) Australia, (**B**) the USA, (**C**) Canada, and (**D**) the UK.

**Figure 3 foods-10-01071-f003:**
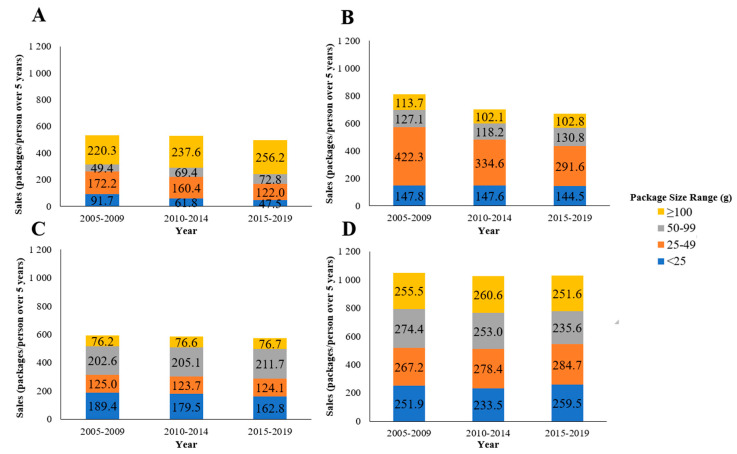
Total package sales in confectionery (retail only) per capita over five-year periods from 2005 to 2019 according to package size bands in (**A**) Australia, (**B**) the USA, (**C**) Canada, and (**D**) the UK.

**Figure 4 foods-10-01071-f004:**
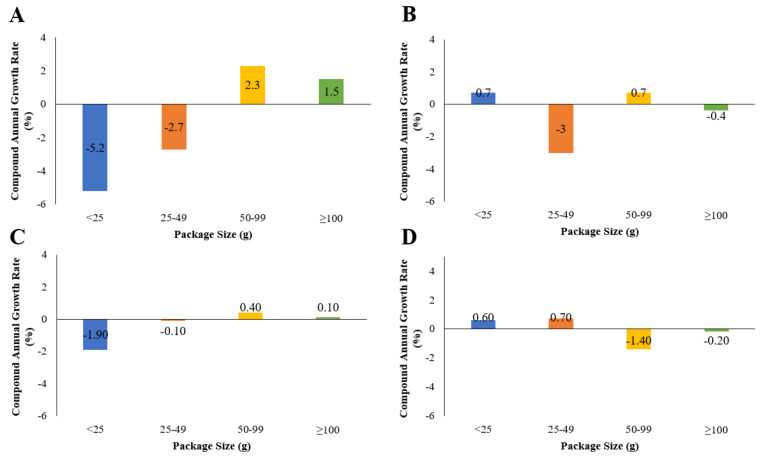
Compound annual growth rate of total package per capita sales of confectionery (retail only) for 2005 to 2019 according to package size band in (**A**) Australia, (**B**) the USA, (**C**) Canada, and (**D**) the UK.

**Figure 5 foods-10-01071-f005:**
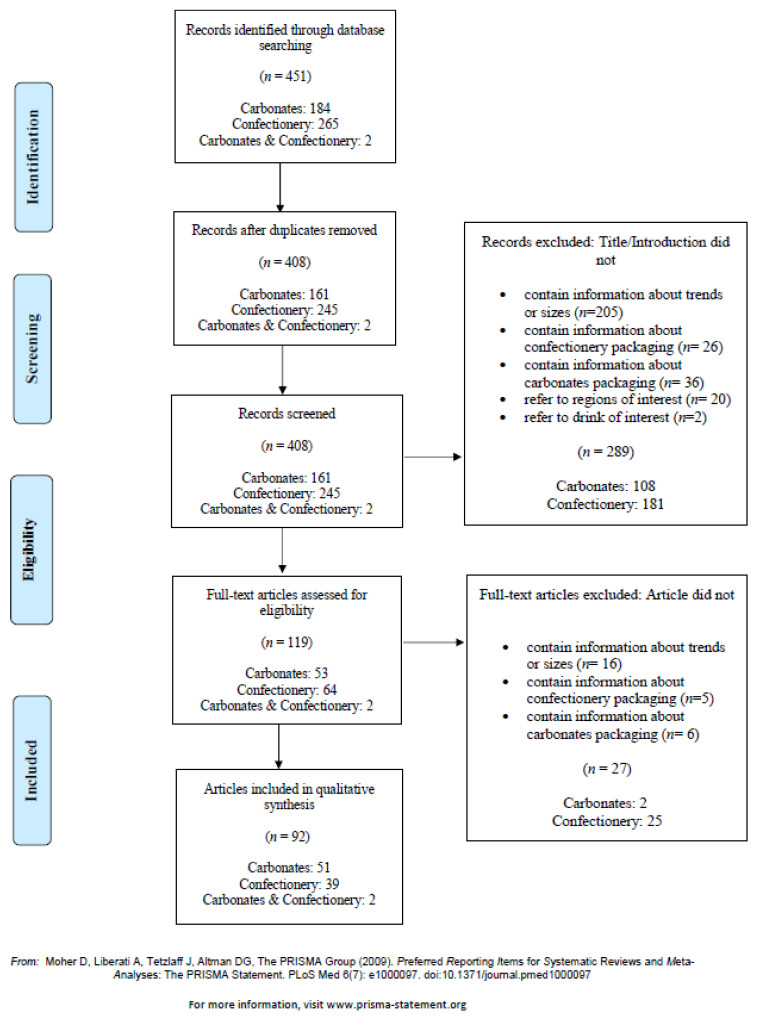
PRISMA flow chart of articles on carbonates and confectionery included in the scoping review.

**Table 1 foods-10-01071-t001:** Industry reports on changes in carbonates package sizes.

	Increase in Package Size	Decrease in Package Size
Country ^1^		
Australia	4	12
USA	3	27
UK	2	19
Canada	2	17
Year		
2005–2009	2	4
2010–2014	2	14
2015–2020	3	3
Reason		
Health consciousness	0	30
Convenience	0	17
Portion control	0	14
Market growth	1	13
Innovation	0	12
Expanding product line	2	11
Value for money	4	9
Profit	0	8
Small indulgence	0	7
Sugar tax	0	6
Premiumisation	0	4
Impulse buying	0	3
Manufacturing costs	0	3
Age group	1	3
Sharing trends	0	1

^1^ Fourteen articles related to multiple countries.

**Table 2 foods-10-01071-t002:** Industry reports on changes in confectionery package size.

	Increase in Package Size	Decrease in Package Size
Country ^1^		
Australia	2	9
USA	8	11
UK	11	26
Canada	6	10
Year		
2005–2009	0	6
2010–2014	8	5
2015–2020	8	24
Reason		
Sharing trends	14	8
Portion control	2	18
Health consciousness	0	16
Small indulgence	1	14
Value for money/affordability	7	8
Manufacturing costs	0	10
Convenience	4	5
Expanding product line	2	6
Profit	2	4
Innovation	2	3
Premiumisation	0	2
Impulse buying	0	1

^1^ Ten articles related to multiple countries.

## Data Availability

Not applicable.

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
