# Peer review of "Trends in Sales and Industry Perspectives of Package Sizes of Carbonates and Confectionery Products"

_foods, 2021, doi:10.3390/foods10051071_

Round 1
Reviewer 1 Report
This is a good quality paper. It is based on reliable secondary data and industry reports. It significantly advances our knowledge about trends in package sizes of carbonates and confectionary products in 4 countries: USA, UK, Canada, and Australia.
line 46 - remove "in Canada"
259 - factors need
344 - remove the duplicate DOI
443 - add DOI
Author Response
Reviewer 1 responses
This is a good quality paper. It is based on reliable secondary data and industry reports. It significantly advances our knowledge about trends in package sizes of carbonates and confectionary products in 4 countries: USA, UK, Canada, and Australia.
Authors response: Thank you so much for your review.
line 46 - remove "in Canada"
Authors response: This has been removed. (line 48)
259 - factors need
Authors response: The text has been amended. (line 276)
344 - remove the duplicate DOI
Authors response: The duplicate DOI has been removed. (line 451)
443 - add DOI line
Authors response: the DOI has been added. (line 560)
Reviewer 2 Report
Overall, this is an interesting analysis because actual data on packages sizes are typically not available for analysis. It’s also useful to see the comparisons across countries for the categories examined. I have some suggestions for improvement but also would encourage the authors in the future to go beyond simple data summaries and attempt to model what factors are associating with changes in package size using regression analyses.
Abstract: Please make is clearer that the articles reviewed were primarily market reports in the trade press rather than peer-reviewed journal articles. Also, the meaning of “Joanna Brigg’s methodology is unknown to this reviewer and likely to others as well.
Line 67: I recommend including a couple of sentences explaining how the results can be used and by whom. Could it help inform policies, the design of voluntary industry initiatives, nutrition education, etc?
Line 71: More information about how Euromonitor acquires its data would be helpful. In particular, do they use store scanner data or manufacturer shipments data? The analysis described in the paper could have been done almost entirely using store scanner data, except for the foodservice portion of carbonates, so it would be useful to understand how scanner data contrasts with Euromonitor’s sources.
Line 74-75: How is package size assessed for foodservice beverages? In many establishments in the U.S., beverages in foodservice are self-serve, so is it based on the number of each cup size sold? What if it’s in a restaurant were drinks are refilled for free?
Line 93-94: Using such wide bands for the package sizes helps with presenting results in the bar charts, but you lose the subtle reductions in package sizes that some manufacturers are doing over time. It’s worth pointing this out as a limitation of your analysis later in the paper. Regression analyses might be more informative in the future so that the more subtle changes can be detected.
Line 110-115: Please make it much more explicit that the search was for market reports and trade publications, not peer-reviewed literature.
Line 133: Indicate in the heading that carbonates include both retail and foodservice.
Figure 1: The scale on the vertical axes is not clear. Does 1 unit mean 1 package? What does it mean if it’s foodservice where beverages are not sold in packages? In the U.S. figure, it looks like the total is about 1300 units, which would mean each person in the U.S. was consuming 3.5 carbonate beverages per day. This doesn’t seem correct, so it’s not clear to mean what a unit is. Also, indicate in the figure title that it includes both retail and foodservice sales volumes.
Line 147-151: The discussion for just retail carbonate beverage sales seems to brief. It could be useful to compare and contrast retail and foodservice.
Line 152: Indicate in the heading that confectionery includes only retail sales.
Figure 3: Here the scale also seems confusing. Is it really the case that people in the U.S. eat at least two retail confectionery products per day? Here as well, indicate in the figure label that it only includes retail.
Line 168: Here as well indicate that the articles were mainly market reports and trade publications, or whatever best describes them. It would be useful to mention examples of names of publications in the text particularly if there were some that were particularly common.
Lines 232-249: It would be useful to cite that reductions in package size in the U.S. for caloric drinks are being driven in part by the Balance Calories Initiative. Also, information on sugar-sweetened beverage taxes could be mentioned along with the results of studies evaluating the effects of these initiatives (see the Healthy Food America website for a lot of detailed information).
Line 258-260: Note that one of the goals of the Balance Calories Initiative is to reduce package sizes. The American Beverage Association produces a publicly available annual report evaluating progress of the initiative that might provide useful insights.
Line 313-315: It seems like this suggests it’s important to look at the total diet, not just individual foods, to assess potential unintended consequences. The paper seems to end somewhat abruptly but it would be useful to discuss implications of the research for policy, nutrition education, etc.
Author Response
Reviewer 2 response
Overall, this is an interesting analysis because actual data on packages sizes are typically not available for analysis. It’s also useful to see the comparisons across countries for the categories examined. I have some suggestions for improvement but also would encourage the authors in the future to go beyond simple data summaries and attempt to model what factors are associating with changes in package size using regression analyses.
Authors response: Thank you so much for your comments and review of this manuscript.
Abstract: Please make is clearer that the articles reviewed were primarily market reports in the trade press rather than peer-reviewed journal articles. Also, the meaning of “Joanna Brigg’s methodology is unknown to this reviewer and likely to others as well.
Authors response: We have added in market research reports and removed the Joanna Briggs methodology from the abstract. (lines 14-15)
Line 67: I recommend including a couple of sentences explaining how the results can be used and by whom. Could it help inform policies, the design of voluntary industry initiatives, nutrition education, etc?
Authors response: We have included a section on how the results can be used: Quantifying how package sizes of unhealthy food and beverages have changed over time helps to develop a better understanding the food environment and how the food industry can influenced the populations diet. Determining the principals that motivate industry to make changes to package sizes will assist in understanding the barriers and enablers of influencing package size reduction and inflation. (lines 70-75)
Line 71: More information about how Euromonitor acquires its data would be helpful. In particular, do they use store scanner data or manufacturer shipments data? The analysis described in the paper could have been done almost entirely using store scanner data, except for the foodservice portion of carbonates, so it would be useful to understand how scanner data contrasts with Euromonitor’s sources.
Authors response: As stated on their website, Euromonitor undertakes primary and secondary data collection sourced from trade interviews, economic indicators, publicly available information, trade news, and company reports. Statistical data on packaged food sources in Euromonitor is obtained from official sources, trade associations, trade press, and company sources, and is available at company and brand level. Additionally, Euromonitor declares that ‘Unlike scan data we cover all distribution channels, including direct sales and informal channels like outdoor markets, while audit data may be confined to only some distribution channels.’ However, the exact sources of the data are not available and therefore the reliability and accuracy are unknown.
We have now added more detail to the Methods section. (lines 85-86)
Line 74-75: How is package size assessed for foodservice beverages? In many establishments in the U.S., beverages in foodservice are self-serve, so is it based on the number of each cup size sold? What if it’s in a restaurant were drinks are refilled for free?
Authors response: We have now added the Euromonitor definition of ‘food service’ to the paper (Table S2); Sales to, not through, foodservice establishments. Includes full- service restaurants, cafés/bars, fast food outlets, 100% home delivery/takeaway, self-service cafeterias, street stalls/kiosks etc. Excludes sales to/through hotels, duty-free sales and institutional sales (sales through/to hospitals, prisons/jails, military, schools, etc, also known as contract foodservice).
Line 93-94: Using such wide bands for the package sizes helps with presenting results in the bar charts, but you lose the subtle reductions in package sizes that some manufacturers are doing over time. It’s worth pointing this out as a limitation of your analysis later in the paper. Regression analyses might be more informative in the future so that the more subtle changes can be detected.
Authors response: The chosen band widths of package sizes were carefully considered, based on providing a range of small, medium and large package sizes, as well as having sufficiently large numbers in each band. Even though we used wide band widths, we were able to detect meaningful changes in package sizes. Using smaller or individual package sizes may show more subtle changes and have added this as a limitation in the Discussion. (line 402-404)
Although regression analysis to determine the predictors of reducing package would be interesting, we feel the available data may not be sufficiently robust to undertake such analysis.
Line 110-115: Please make it much more explicit that the search was for market reports and trade publications, not peer-reviewed literature.
Authors response: This has now been added. (line 123)
Line 133: Indicate in the heading that carbonates include both retail and foodservice.
Authors response: This has now been added. (line 150)
Figure 1: The scale on the vertical axes is not clear. Does 1 unit mean 1 package? What does it mean if it’s foodservice where beverages are not sold in packages? In the U.S. figure, it looks like the total is about 1300 units, which would mean each person in the U.S. was consuming 3.5 carbonate beverages per day. This doesn’t seem correct, so it’s not clear to mean what a unit is. Also, indicate in the figure title that it includes both retail and foodservice sales volumes.
Authors response: Apologies for any confusion. The figures have now been updated; unit has been changed to package; sales are over 5 years, not 1 year. Packages (cans, bottles) are sold to foodservice establishments in a similar manner as in retail. Post-mix carbonates are not included in this analysis.
Line 147-151: The discussion for just retail carbonate beverage sales seems to brief. It could be useful to compare and contrast retail and foodservice.
Authors response: Thank you, we have now expanded this section.
‘However, in foodservice carbonates can be sold in packages (can or bottles) or in cups using post-mix dispensers. The latter measure is unavailable in the Euromonitor database. The most popular package sizes were similar between retail and foodservice in Australia and Canada (300-399 mL) but differed in the US (400-999 mL more popular in foodservice) and the UK (<300 mL more popular in foodservice). Reasons for these differences can only be speculated but likely due to different target population and cultural expectations.’ (line 336-342)
Line 152: Indicate in the heading that confectionery includes only retail sales.
Authors response: This has now been added. (line 185)
Figure 3: Here the scale also seems confusing. Is it really the case that people in the U.S. eat at least two retail confectionery products per day? Here as well, indicate in the figure label that it only includes retail.
Authors response: Apologies for any confusion. The figures have now been updated; unit changed to package; and sales are indicated over 5 years, not 1 year, this has now been clarified.
Line 168: Here as well indicate that the articles were mainly market reports and trade publications, or whatever best describes them. It would be useful to mention examples of names of publications in the text particularly if there were some that were particularly common.
Authors response: This has now been clarified.
‘Articles came from a variety of sources such as ConfectioneryNews.com, Forbes, Beveragedaily.com, Euromonitor research reports, and direct from company websites.’ (lines 215-217)
Lines 232-249: It would be useful to cite that reductions in package size in the U.S. for caloric drinks are being driven in part by the Balance Calories Initiative. Also, information on sugar-sweetened beverage taxes could be mentioned along with the results of studies evaluating the effects of these initiatives (see the Healthy Food America website for a lot of detailed information).
Authors response: Thank you for this suggestion, we have now added some details on the voluntary industry initiative Balance Calories Initiative and its interim evaluation reports.
‘Voluntary industry initiatives such as the Balance Calories Initiative comprising the top three American beverage companies have committed to promote the use of smaller package sizes. An interim 2020 report found some growth among smaller containers of caloric beverages but this was offset by shifts from medium-sized to larger containers (Keybridge 2020).’ (lines 350-355)
Keybridge. 2025 Beverage Calories Initiative: Report on 2019 Progress toward the National Calorie Goal. September 2020. Available online: https://aba-bigtree.s3.amazonaws.com/files/resources/bic-2019-national-progress-report.pdf (accessed on 3 May 2021)
In the USA there are no states with an excise tax on sugar-sweetened beverages; only local levies are in place in selected cities/districts: Boulder, Colorado; the District of Columbia; Philadelphia, Pennsylvania; Seattle, Washington; and four California cities: Albany, Berkeley, Oakland, and San Francisco. This has now been added to the Discussion (lines 330-332).
Line 258-260: Note that one of the goals of the Balance Calories Initiative is to reduce package sizes. The American Beverage Association produces a publicly available annual report evaluating progress of the initiative that might provide useful insights.
Authors response: Thank you for this suggestion, we have now added more details, see response above. (lines 350-355)
Line 313-315: It seems like this suggests it’s important to look at the total diet, not just individual foods, to assess potential unintended consequences. The paper seems to end somewhat abruptly but it would be useful to discuss implications of the research for policy, nutrition education, etc.
Authors response: Thank you, we have now elaborated further:
‘Promisingly, health consciousness is the primary driver for reductions in both confectionary and SSB. However, industry is also motivated to increase profitability and employs a variety of strategies such as using both larger and smaller package sizes to increase sales to consumers who are motivated by factors other than health such as value for money. As smaller packages also drive sales and potentially allow new indulges that may not have occurred with larger sizes perceived to be unhealthy, further research into potential unintended risks associated with consumption of smaller package sizes are warranted.’ (lines 411-421).
Reviewer 3 Report
It was with great pleasure that I have reviewed the study by Jensen et al. which aimed to determine sales trends, and reported food industry perspectives for changing food package sizes of carbonates and confectionery between 2005 and 2019. The manuscript is well structured and well-written, presenting relevant results on the field and an adequate discussion. I have only minor modifications to ask the authors:
The Introduction is quite superficial. A more in-depth approach is needed, namely on the legal framework and more information on the consumption of carbonates and confectionery in the countries that the study proposes to analyze.
Lines 66-67: Please specify the countries.
It would be useful to add a flowchart in section 2 with all the steps taken in conducting the present study.
Figures 1, 2, 3 and 4 need to be improved. Some of the data are difficult to read.
Study limitations should be highlighted at the end of Discussion section.
Author Response
Reviewer 3 response
It was with great pleasure that I have reviewed the study by Jensen et al. which aimed to determine sales trends, and reported food industry perspectives for changing food package sizes of carbonates and confectionery between 2005 and 2019. The manuscript is well structured and well-written, presenting relevant results on the field and an adequate discussion. I have only minor modifications to ask the authors:
Authors response: Thank you very much for your review and comments to improve the manuscript.
The Introduction is quite superficial. A more in-depth approach is needed, namely on the legal framework and more information on the consumption of carbonates and confectionery in the countries that the study proposes to analyze.
Authors response: Thank you for these suggestions. There are no restrictions on the amount of product that can be placed in packages, and the only laws governing package size relate to "slack fill" in some countries. Slack fill is not relevant to our study aims, as we were interested in the portion weight of the foods and beverages, not the external appearance of the package. Accordingly, we used the gram weight of confectionary or the mLs of beverages. It is a legal requirement that the net weight of food is provided on all packaged foods in all four countries and this is now stated at lines 70-71. The main points of interest regarding consumption of confectionary and carbonates are presented at lines 41-49.
Lines 66-67: Please specify the countries.
Authors response: These have now been added. (lines 67-68)
It would be useful to add a flowchart in section 2 with all the steps taken in conducting the present study.
Authors response: The steps undertaken to conduct the search strategy used to extract sales data and industry reports are described in Supplementary materials (Figure S1 and Table S1).
Figures 1, 2, 3 and 4 need to be improved. Some of the data are difficult to read.
Authors response: The figures have now been updated and the font size increased.
Study limitations should be highlighted at the end of Discussion section.
Authors response: The study limitations have now been highlighted. (lines 397-404)
Round 2
Reviewer 2 Report
My comments on the first draft have been adequately addressed--thank you.